# Efficacy and safety of tranexamic acid in intracranial haemorrhage: A meta-analysis

**Yu Xiong**[1☯], **Xiumei Guo**[1,2☯], **Xinyue Huang**[1☯], **Xiaodong Kang**[1☯], **Jianfeng Zhou**[1], **Chunhui Chen**[1], **Zhigang Pan**[1], **Linxing Wang**[2], **Roland Goldbrunner**[3], **Lampis Stavrinou**[4], **Pantelis Stavrinou**[3,5], **Shu Lin**[6,7]*, **Yuping Chen**[1]*, **Weipeng Hu**[1]*, **Feng Zheng**[1]*

1 Department of Neurosurgery, The Second Affiliated Hospital, Fujian Medical University, Quanzhou, Fujian Province, China, 2 Department of Neurology, The Second Affiliated Hospital, Fujian Medical University, Quanzhou, Fujian Province, China, 3 Department of Neurosurgery, Center for Neurosurgery, Faculty of Medicine and University Hospital, University of Cologne, Cologne, Germany, 4 2nd Department of Neurosurgery, "Attikon" University Hospital, National and Kapodistrian University, Athens Medical School, Athens, Greece, 5 Neurosurgery, Metropolitan Hospital, Athens, Greece, 6 Centre of Neurological and Metabolic Research, The Second Affiliated Hospital of Fujian Medical University, Quanzhou, China, 7 Diabetes and Metabolism Division, Garvan Institute of Medical Research, Sydney, NSW, Australia

☯ These authors contributed equally to this work.
* dr.feng.zheng@gmail.com (FZ); neurosurgery_fyey@163.com (WH); 503106356@qq.com (YC); shulin1956@126.com (SL)

**Data Availability Statement:** All relevant data are within the paper and its Supporting Information files.

**Funding:** The author(s) received no specific funding for this work.

## Abstract

### Background

Although some studies have shown that tranexamic acid is beneficial to patients with intracranial haemorrhage, the efficacy and safety of tranexamic acid for intracranial haemorrhage remain controversial.

### Method

The PubMed, EMBASE, and Cochrane Library databases were systematically searched. The review followed PRISMA guidelines. Data were analyzed using the random-effects model.

### Results

Twenty-five randomized controlled trials were included. Tranexamic acid significantly inhibited hematoma growth in intracranial hemorrhage (ICH) and traumatic brain injury (TBI) patients. (ICH: mean difference -1.76, 95%CI -2.78 to -0.79, $I^2$ = 0%, P < .001; TBI: MD -4.82, 95%CI -8.06 to -1.58, $I^2$ = 0%, P = .004). For subarachnoid hemorrhage (SAH) patients, it significantly decreased the risk of hydrocephalus (OR 1.23, 95%CI 1.01 to 1.50, $I^2$ = 0%, P = .04) and rebleeding (OR, 0.52, 95%CI 0.35 to 0.79, $I^2$ = 56% P = .002). There was no significance in modified Rankin Scale, Glasgow Outcome Scale 3–5, mortality, deep vein thrombosis, pulmonary embolism, or ischemic stroke/transient ischemic.

**Competing interests:** The authors have declared that no competing interests exist.

## Conclusion

Tranexamic acid can significantly reduce the risk of intracranial haemorrhage growth in patients with ICH and TBI. Tranexamic acid can reduce the incidence of complications (hydrocephalus, rebleeding) in patients with SAH, which can indirectly improve the quality of life of patients with intracranial haemorrhage.

## 1. Introduction

Intracranial haemorrhage is a neurological disease with high morbidity and mortality [1]. It is a serious form of stroke and a common cause of death and disability [2]. Early hemostasis is key to treating patients with intracranial haemorrhage [3]. Hematoma enlargement is a powerful prognostic factor associated with neurological function in patients with intracranial haemorrhage. Therefore, preventing early hematoma growth is the main goal in the treatment of acute intracranial haemorrhage [4].

Tranexamic acid is a widely used anti-fiber solvent [5]. It is a synthetic lysine analog that competes with lysine residues on fibrin for the binding of plasminogen that effectively inhibits the interaction between fibrinolytic enzymes and fibrin and prevents the dissolution of fibrin clots [6]. In a randomized trial involving 2325 participants, tranexamic acid effectively reduced hematoma expansion, but its hemostatic effect was not sufficient to translate into improved functional outcomes [7]. In addition, the effect of tranexamic acid on improving neurological function in patients with intracranial haemorrhage remains uncertain, and the safety of tranexamic acid in reducing mortality and rebleeding complications in patients with intracranial haemorrhage remains controversial [2, 4, 8–10]. Therefore, we conducted a meta-analysis to evaluate the hemostatic effect of tranexamic acid and its safety in improving neurological function in patients with intracranial haemorrhage.

## 2. Methods

### 2.1 Search strategy

Two researchers (YX and FZ) conducted a comprehensive search of the PubMed, EMBASE, and Cochrane Library databases. All clinical randomized controlled trials (RCTs) published on tranexamic acid in patients with intracranial haemorrhage before September 2022 were included. Keywords used in the retrieval strategy included "tranexamic acid", "intracranial haemorrhage", "intracranial hemorrhage", "subarachnoid hemorrhage", "traumatic hemorrhage", and "randomized controlled study". The keywords used in this search strategy included "Randomized Controlled Trial" AND ("Subarachnoid Hemorrhage" OR "SAH" OR "Subarachnoid Hemorrhage, Aneurysmal" OR "Subarachnoid Hemorrhage, Spontaneous" OR "Perinatal Subarachnoid Hemorrhage"OR "Subarachnoid Hemorrhage, Intracranial" OR "Brain Injuries, Traumatic" OR "Trauma, Brain" OR "TBI" OR "Encephalopathy, Traumatic" OR "Posterior Fossa Hemorrhage" OR "Brain Hemorrhage" OR "Intracranial Hemorrhages") AND ("AMCHA" OR "trans-4" OR "t-AMCHA" OR "AMCA" OR "Anvitoff" OR "Cyklokapron" OR "Ugurol" OR "KABI 2161" OR "Spotof" OR "Transamin" OR "Amchafibrin" OR "Exacyl" OR "Tranexamic Acid"). The medical keywords database was used to identify synonyms. Repetitive literature, case reports, conference summaries, animal experiments, experimental designs, and ongoing experiments were excluded from the analysis. Before data

analysis, literature retrieval was performed again to ensure that newly published eligible literature was included. The review was not registered.

## 2.2 Inclusion and exclusion criteria

The inclusion criteria of the eligible studies were as follows: (1) patients with intracranial haemorrhage diagnosed by computed tomography or magnetic resonance imaging, including intracranial hemorrhage (ICH), subarachnoid hemorrhage (SAH), and traumatic brain injury (TBI); (2) intervention was tranexamic acid therapy (regardless of dose and injection time); the controls employed a matching placebo or 0.9% saline according to the same administration schedule; (3) outcome indicators included: modified Rankin scale (mRS), Glasgow Outcome Scale (GOS), hematoma growth, hematoma expansion, and safety results, including mortality, hydrocephalus, deep vein thrombosis (DVT), pulmonary embolism (PE), seizures, ischemic stroke (IS) or transient ischemic attack (TIA), and rebleeding; and (4) the study was a clinical RCT.

Additionally, the exclusion criteria were as follows: (1) interventions were non-tranexamic acid treatment; (2) other diseases; (3) compared tranexamic acid with other blood-clotting agents; (4) non-randomized controlled studies such as retrospective studies, case reports, reviews, meetings, and letters; (5) ongoing studies; and (6) studies that were still in the early design stage.

## 2.3 Selection of studies

Two researchers (YX and FZ) selected studies that met the inclusion criteria according to the research title and abstract. The full text of the preliminary study was then retrieved to assess whether they qualified for inclusion. For multiple results reported at different timepoints, we used the last result reported at the longest follow-up time point. Discrepancies were resolved through discussion.

## 2.4 Data extraction

A researcher (YX) used a pre-designed standardized form to extract data, and the data were checked by a second researcher (FZ) to evaluate the risk of bias and evidence quality of individual included studies. The extracted information included the type of study, type of intracranial haemorrhage, sex ratio of subjects, details of interventions, outcome indicators, and other data used to assess the risk of bias and quality of evidence. Whenever possible, missing data were obtained via email from the study authors.

## 2.5 Quality of assessment

The Cochrane Risk Bias Assessment Tool was used to assess bias in each study. The degree of bias risk (low, unknown, high) was evaluated from seven aspects (random sequence generation, allocation concealment, blinding of researchers and subjects, integrity of outcome data, blind evaluation of research outcomes, selective reporting of research results, and other sources of bias) to reflect the quality of each study.

## 2.6 Outcome measures

The outcome measures included hemostatic effects (hematoma growth, hematoma expansion), efficacy outcomes (mRS and GOS), mortality, and safety outcomes (DVT, PE, IS or TIA, hydrocephalus, seizures, and rebleeding). Hematoma expansion was defined as an increase of >6 mL or >33% in hematoma volume on 24-h computed tomography scans compared with

the baseline hematoma volume [11]. The mRS and GOS are mostly used to evaluate the neurological function of patients with intracranial haemorrhage. An mRS score ≤2 is considered to have a good prognosis, generally does not leave a disability after treatment, and does not affect the quality of daily life of patients [12]. A GOS score 3–5 is generally considered a conscious state that does not threaten the safety of life [13].

## 2.7 Statistical analysis

The meta-analysis of data was performed using REVMAN software (version 5.3 for Windows; Cochrane Collaboration, Copenhagen, Denmark). We obtained 95% confidence intervals (CIs) and P-values from each study by measuring the mean difference (MD) of continuous variables and the odds ratio (OR) of binary variables. Q and $I^2$ statistics were used to test the heterogeneity of the studies. According to the recommendations of the Cochrane Statistical Methods Group, the heterogeneity P-value was set to 0.1 and the $I^2$ statistic was interpreted as follows: 0–40%, low heterogeneity; 30–60%, moderate heterogeneity; 50–90%, substantive heterogeneity; 75–100%, obvious heterogeneity. Sensitivity analysis was performed to identify the source of heterogeneity if the heterogeneity was substantial ($I^2 \geq 50\%$) [14].

For outcome indicators of more than 10 studies, researchers drew a funnel plot (RevMan 5.3) to evaluate publication bias [15]. We used Egger's test and Begg's test (STATAMP-64) to evaluate the asymmetry of the funnel plot, and $P > .05$ was considered as no publication bias.

## 3. Results

### 3.1 Results of the search

A total of 207 studies were retrieved and 183 articles were included after removing duplicate articles. Another 114 articles were excluded during primary screening (intervention without tranexamic acid, study of disease without intracranial haemorrhage, meta-analysis and review, conference summary, and animal experiments). After a full review of the remaining 69 studies, 29 ongoing studies and 13 studies that did not meet the inclusion criteria were excluded. The retrieval strategy was repeated before data analysis, and no additional studies were identified in the update. Finally, 25 studies involving 20146 patients were included. A flow chart of the specific search strategy results is shown in S1 Fig. The overall characteristics of the 25 studies and the specific information on the individual studies are shown in Table 1.

### 3.2 Risk of bias in included studies

The Cochrane Risk Bias Assessment Tool was used to assess the quality of the included 27 RCTs. Among them, 26 studies [3, 7, 11, 12, 16–20, 22–28, 30–32, 38] mentioned generating random sequences, including through random number tables, computer random number generation, coin-throwing, and double-blind lottery. One study [21] did not specify the randomization process. Nineteen studies [3, 7, 11, 12, 17, 18, 27, 31, 38] were blinded for both subjects and researchers, and one study [16] was only single-blinded. One study [25] was non-blinded to the researchers or subjects (open label), and six studies [20–22, 24, 26, 28] did not elaborate on blindness. The outcome measurement of one study [31] was completed by a research assistant who knew the purpose and conditions of the study, while two studies [21, 24] did not mention whether they conducted a blind evaluation of the outcome. In one study [17], two patients who took tranexamic acid were lost to follow-up, and there were no 24-h follow-up data. The results of the bias risk assessment are shown in S2 Fig.

**Table 1. The characteristics of the 29 included studies.**

| Author | Year | Design | Type of bleed | TXA | placebo | Enrollment time after bleed | TXA dose | Length of follow-up | Outcome |
|---|---|---|---|---|---|---|---|---|---|
| Arumugam [16] | 2015 | RCT | ICH | 15 | 15 | Within 8 h | 2g/day | 1 day | HG |
| Liu [17] | 2021 | RCT | ICH | 89 | 82 | Within 8 h | 2g/day | 3 months | HG mRS Mortality DVT Seizures |
| Meretoja [3] | 2020 | RCT | ICH | 50 | 50 | within 4.5 h | 2g/day | 3 months | mRS Mortality IS or TIA PE |
| Sprigg [18] | 2014 | RCT | ICH | 16 | 8 | Within 24 h | 2g/day | 3 months | HG mRS Mortality DVT IS or TIA |
| Sprigg [7] | 2018 | RCT | ICH | 1161 | 1164 | Within 8 h | 2g/day | 3 months | HG mRS Mortality |
| Chandra [19] | 1978 | RCT | SAH | 20 | 19 | Within 7 days | 6g/day | 21 days | Mortality |
| Fodstad [20] | 1981 | RCT | SAH | 30 | 29 | Within 72 h | 6g/day | 42 days | Mortality IS or TIA rebleeding |
| Gibbs [21] | 1971 | RCT | SAH | 25 | 22 | NR | 3g/day | 2 months | Mortality rebleeding |
| Hillman [22] | 2002 | RCT | SAH | 254 | 251 | Within 48 h | 4g/day | 6 months | GOS rebleeding |
| Kaste [23] | 1979 | RCT | SAH | 32 | 32 | Within 72 h | 6g/day | 14 days | Mortality rebleeding |
| Maurice [24] | 1978 | RCT | SAH | 25 | 25 | Within 96 h | 6g/day | 3–33 months | Mortality hydrocephalus |
| Post [25] | 2021 | RCT | SAH | 480 | 475 | Within 24 h | 3g/day | 6 months | mRS Mortality DVT Seizures Hydrocephalus IS or TIA PE rebleeding |
| Roos [26] | 2000 | RCT | SAH | 229 | 233 | Within 96 h | 6g/day | 3 months | GOS Hydrocephalus IS or TIA rebleeding |
| Tsementzis [27] | 1990 | RCT | SAH | 50 | 50 | Within 72 h | 9g/day | 6 months | GOS Mortality rebleeding |
| Vermeulen [28] | 1984 | RCT | SAH | 241 | 238 | Within 72 h | 6g/day | 3 months | GOS Mortality DVT Hydrocephalus IS or TIA PE rebleeding |
| CRASH 2 [29] | 2012 | RCT | TBI | 133 | 137 | Within 8 h | 2g/day | 28 days | HG IS or TIA rebleeding |
| CRASH-3 [30] | 2019 | RCT | TBI | 6359 | 6280 | Within 3 h | 2g/day | 28 days | DVT Seizures IS or TIA PE |
| Chakroun [13] | 2018 | RCT | TBI | 96 | 84 | Within 24 h | 2g/day | 28 days | GOS Mortality DVT PE rebleeding |
| Ebrahimi [31] | 2019 | RCT | TBI | 40 | 40 | Within 8 h | 2g/day | 7 days | Mortality |
| Fakharian [32] | 2018 | RCT | TBI | 74 | 75 | Within 8 h | 1g/day | 3 months | GOS Mortality rebleeding |
| Jokar [33] | 2017 | RCT | TBI | 40 | 40 | Within 2 h | 2g/day | 2 days | HG |
| Mousavinejad [34] | 2020 | RCT | TBI | 20 | 20 | Within 8 h | 2g/day | 6h | Mortality |
| Rowell [35] | 2020 | RCT | TBI | 657 | 309 | Within 2 h | 2g/day | 6 months | Mortality DVT Seizures IS or TIA PE |
| Safari [36] | 2021 | RCT | TBI | 47 | 47 | NR | 4g/day | 2 days | HG |
| Yutthakasemsunt [37] | 2013 | RCT | TBI | 120 | 118 | Within 8 h | 2g/day | 1 days | Mortality IS or TIA |

ICH: intracranial hemorrhage; TBI: traumatic brain injury; SAH: subarachnoid hemorrhage; HG: hematoma growth; IS: ischemic stroke; mRS: modified Rankin Scale; DVT: deep vein thrombosis; PE: deep vein thrombosis; TIA: transient ischemic attack; GOS: Glasgow Outcome Scale; RCT: randomized controlled trial; TXA:transamic acid.

## 3.3 Outcomes

**3.3.1 ICH.** *3.3.1.1 Hematoma growth.* Four studies [7, 16–18] reported hematoma growth at the end of follow-up, and the pooled data showed that the hematoma growth rate of the tranexamic acid intervention group was lower than that of the control group (MD -1.78, 95%CI -2.78 to -0.79, $I^2$ = 0%, P < .001; Fig 1A) for ICH patients.

*3.3.1.2 mRS.* Three studies [3, 17, 18] reported mRS scores at the end of follow-up, and pooled data showed that tranexamic acid had no significant benefit on the prognosis of patients with ICH (MD 0.16, 95%CI -0.05 to 0.37, $I^2$ = 0%, P = .13; Fig 1B).

*3.3.1.3 mRS≤2.* Three studies [3, 7, 17] reported an mRS score≤2 (non-disabling stroke patients) at the end of follow-up. No significant difference was detected between the tranexamic acid and control groups in ICH patients (OR 1.05, 95%CI 0.89 to 1.24, $I^2$ = 0%; P = .58; Fig 1C).

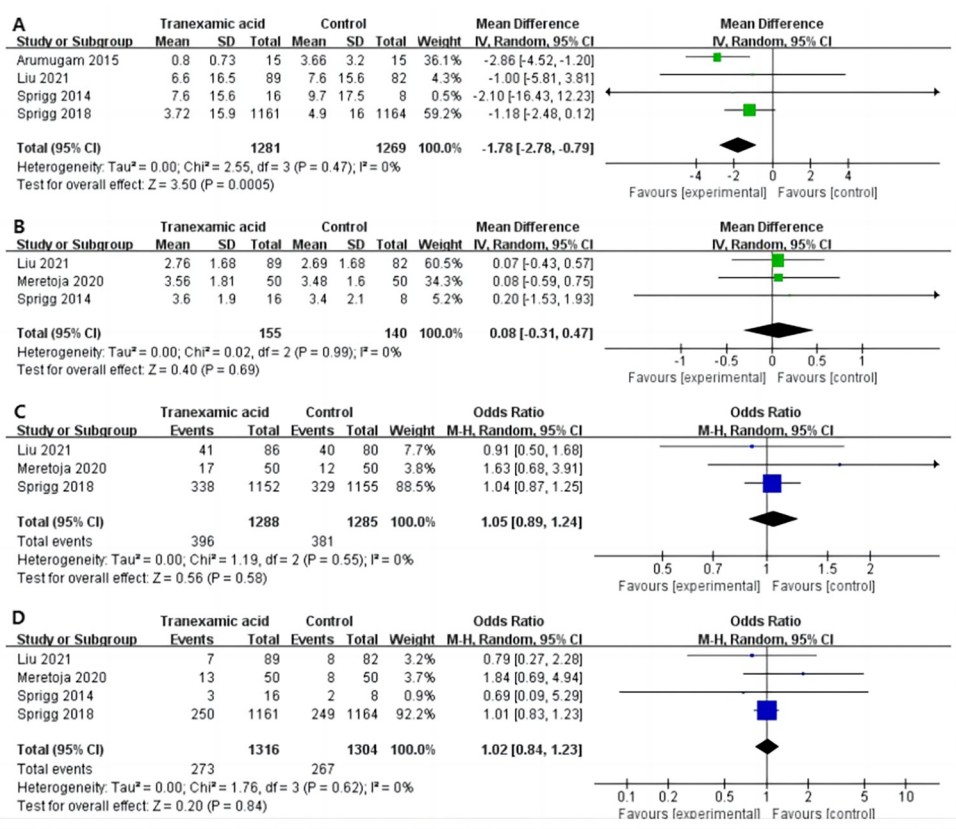

**Fig 1.** Forest plot of the effect of tranexamic acid on hematoma growth (A), mRS (B), mRS≤2 (C), and mortality (D) in patients with ICH.

*3.3.1.4 Mortality.* Four studies [3, 7, 17, 18] reported the number of deaths by the end of follow-up. There was no significant difference in the incidence of deaths between the tranexamic acid and control groups in ICH patients (OR 1.02, 95%CI 0.84 to 1.23, $I^2$ = 0%, P = .84; Fig 1D).

**3.3.2 SAH.** *3.3.2.1 GOS 3–5.* Four studies [22, 26–28] reported a GOS score of 3–5 at the end of follow-up. There was no significant difference between the tranexamic acid intervention and control groups in SAH patients (OR 1.13, 95%CI 0.91 to 1.41, $I^2$ = 0%, P = .26; Fig 2A).

*3.3.2.2 Mortality.* Eight studies [19–21, 23–25, 27, 28] reported the number of deaths by the end of follow-up. There was no significant difference in the incidence of deaths between the tranexamic acid and control groups in SAH patients (OR 0.84, 95%CI 0.54 to 1.31, $I^2$ = 57%, P = .83; Fig 2B).

*3.3.2.3 DVT.* Three studies [19, 25, 28] reported the occurrence of DVT at the end of follow-up. No significant difference was detected between the tranexamic acid intervention group and the control group in SAH patients (OR 1.08, 95%CI 0.51 to 2.30, $I^2$ = 0%, P = .84; Fig 2C).

*3.3.2.4 IS or TIA.* The occurrence of IS or TIA was reported in four studies [20, 25, 26, 28] at the end of follow-up. No significant difference was detected between the tranexamic acid and control groups in SAH patients (OR 1.20, 95%CI 0.83 to 1.72, $I^2$ = 56%, P = .33; Fig 2D).

*3.3.2.5 Rebleeding.* Rebleeding was reported in nine studies [19–23, 25–28] at the end of follow-up, and the data showed that the incidence of rebleeding in the tranexamic acid group was

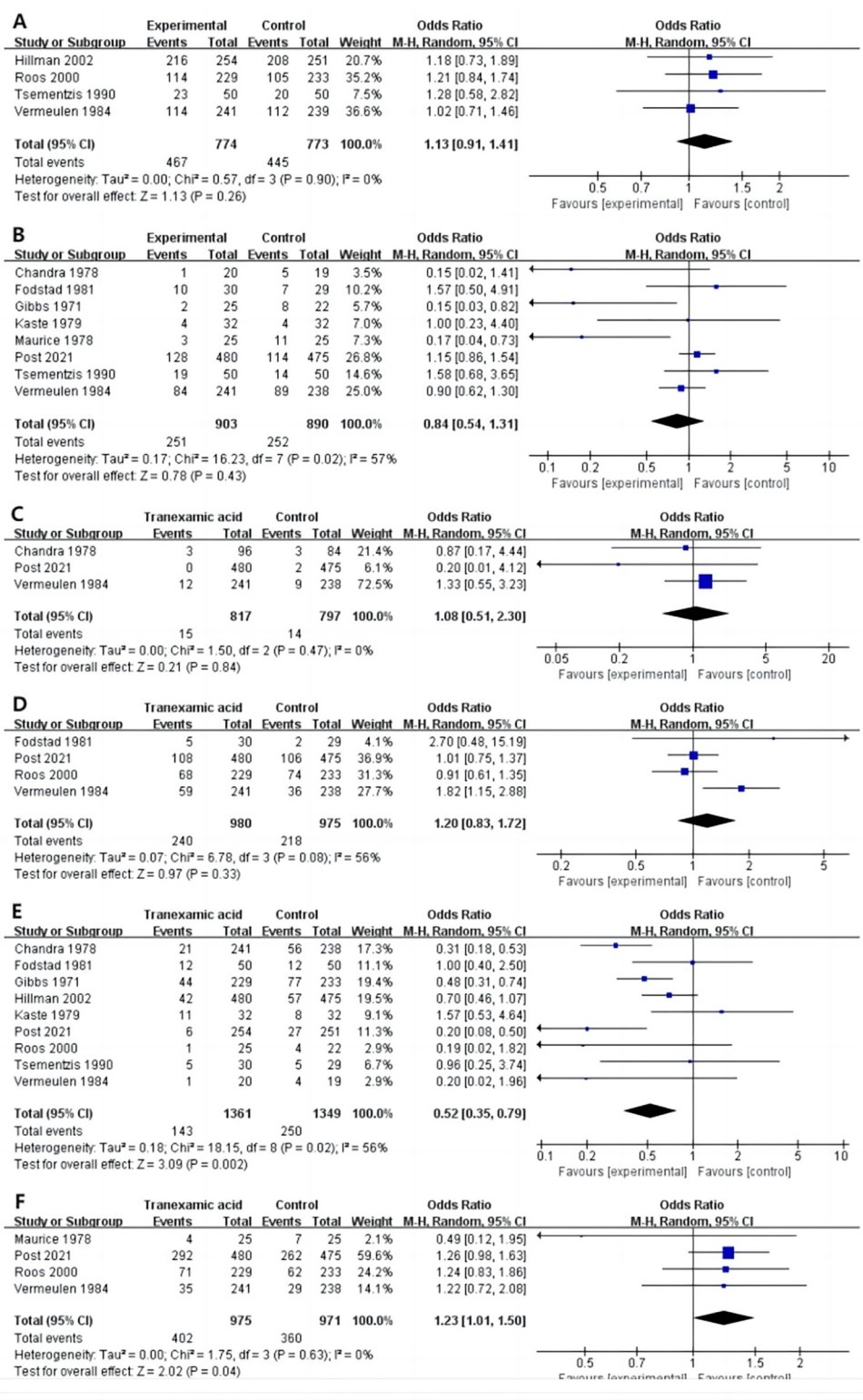

**Fig 2.** Forest plot of the effect of tranexamic acid on GOS 3–5 (A), Mortality (B), DVT (C), IS or TIA (D), and Rebleeding (E) in patients with SAH.

lower than that in the control group in SAH patients (OR 0.52, 95%CI 0.35 to 0.79; $I^2$ = 56%, *P* = .002; Fig 2E).

*3.3.2.6 Hydrocephalus.* Four studies [24–26, 28] reported hydrocephalus at the end of follow-up, and pooled data showed that the incidence of hydrocephalus in the tranexamic acid group was lower than that in the control group in SAH patients (OR 1.23, 95%CI 1.01 to 1.50; $I^2$ = 0%, P = .04; Fig 2F).

**3.3.3 TBI.** *3.3.3.1 Hematoma growth.* Three studies [29, 33, 36] reported hematoma growth at the end of follow-up, and the pooled data showed that the hematoma growth rate of the tranexamic acid intervention group was lower than that of the control group (MD -4.86, 95%CI -8.06 to -1.58, $I^2$ = 0%, P = .004; Fig 3A) for TBI patients.

*3.3.3.2 Mortality.* Six studies [13, 31, 32, 34, 35, 37] reported the number of deaths by the end of follow-up. There was no significant difference in the incidence of deaths between the tranexamic acid and control groups in TBI patients (OR 0.91, 95%CI 0.69 to 1.21, $I^2$ = 0%, P = .52; Fig 3B).

*3.3.3.3 PE.* Three studies [13, 30, 35] reported the occurrence of PE at the end of follow-up. No significant difference was detected between the tranexamic acid intervention and control groups in TBI patients (OR 1.22, 95%CI 0.45 to 3.27, $I^2$ = 65%; P = .70; Fig 3C).

*3.3.3.4 IS or TIA.* The occurrence of IS or TIA was reported in four studies [29, 30, 35, 37] at the end of follow-up. No significant difference was detected between the tranexamic acid and control groups in TBI patients (OR 0.81, 95%CI 0.51 to 1.30, $I^2$ = 23%, P = .39; Fig 3D).

*3.3.3.5 Seizures.* Two studies [30, 35] reported the occurrence of seizures at the end of the follow-up. There was no significant difference between the tranexamic acid intervention and control groups in TBI patients (OR 1.11, 95%CI 0.92 to 1.36, $I^2$ = 0%, P = .28; Fig 3E).

## 4. Discussion

This meta-analysis involved 25 studies with a total of 20,146 participants, providing evidence to evaluate tranexamic acid from three aspects: hemostatic effect, efficacy in improving neurological function, and complications. Compared with the control group, the tranexamic acid group significantly reduced hematoma growth in ICH and TBI. For SAH patients, tranexamic acid significantly reduced the probability of hydrocephalus and rebleeding. Mortality, IS or TIA, DVT, PE, and risk of seizures were similar between the two groups.

Compared with patients without hematoma growth, those with hematoma growth have an increased risk of neurological deterioration and mortality associated with intracranial haemorrhage [4]. Our data showed that tranexamic acid can inhibit hematoma growth, which is consistent with a previous study [39, 40]. This may be because tranexamic acid is a synthetic lysine derivative, which is an antifibrinolytic agent [41]. Tranexamic acid competitively inhibits the adsorption of plasminogen on fibrin by binding to the lysine binding site on fibrin, thereby inhibiting the activation of plasminogen and preventing the degradation of fibrinolytic proteins by fibrinolytic enzymes, thus achieving antifibrinolytic and hemostatic effects [42, 43]. It can also increase collagen synthesis in fibrin clots, thereby increasing the strength and stability of clots and reducing bleeding [5, 44]. Normally, fibrinogen binds to lysine binding sites on fibrin molecules and is converted to fibrinogenase in the presence of tissue plasminogen activator [45]. TXA blocks activation of plasminogen and prevents binding of plasmin to fibrin [46].

Our results showed that tranexamic acid could significantly reduce the occurrence of hematoma growth in the brain, although no significant effect was detected on the recovery of neurological function (ICH: mRS, P = .69; mRS≤2, P = .58. SAH: GOS 3–5, P = .26) or mortality (ICH: P = .84; SAH: P = 0.43; TBI: P = 0.52). This may be because secondary brain injury is

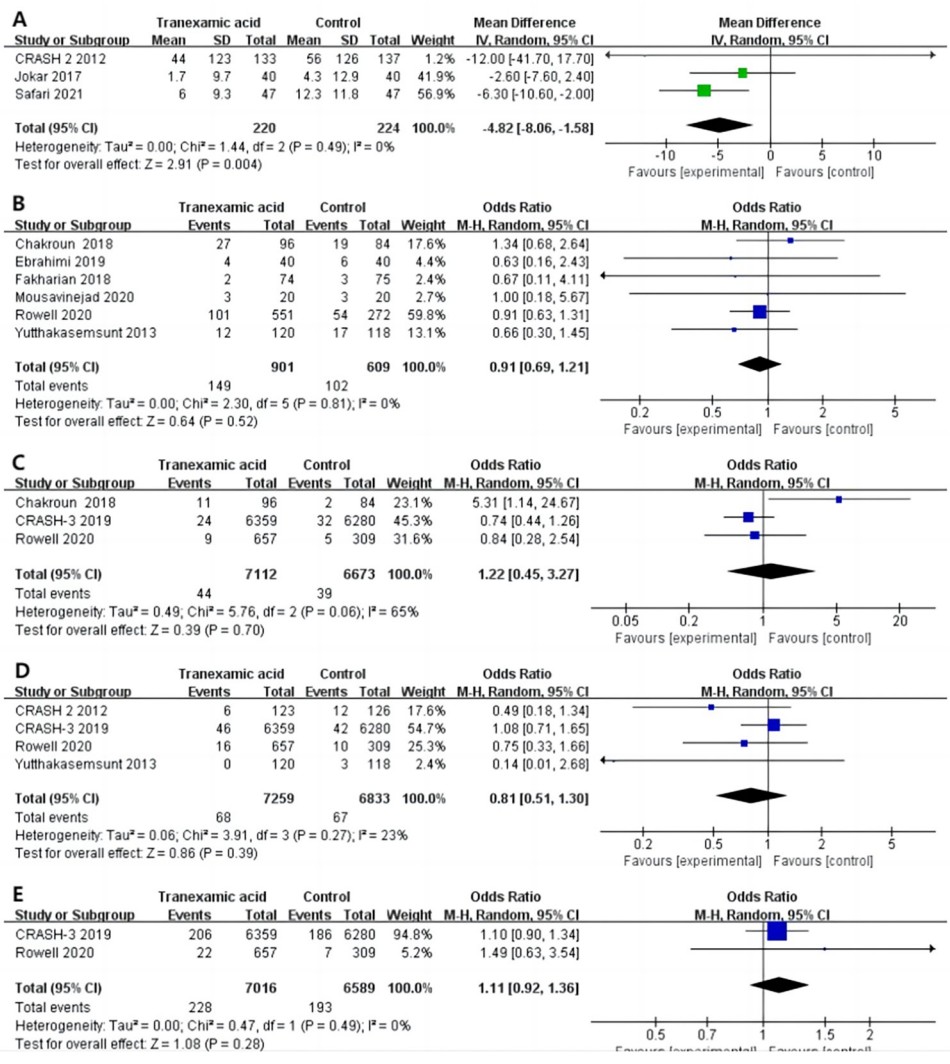

**Fig 3.** Forest plot of the effect of tranexamic acid on Hematoma growth (A), Mortality (B), PE (C), IS or TIA (D), and Seizures (E) in patients with SAH.

caused by inflammation; oxidative stress or cytotoxicity dominates when the hematoma volume is large, in addition to the mechanical compression of the surrounding brain tissue [47]. Therefore, it is difficult to improve neurological function in patients with intracranial haemorrhage. However, the presence or absence of hematoma growth is also associated with blood pressure control, cerebral amyloid angiopathy, chronic kidney disease and lifestyle (smoking, alcohol consumption), which may confuse the reported association [48–50]. Additionally, the antifibrinolytic effect of tranexamic acid can promote the stability of thrombosis, and the effective benefits may be offset by the increase in ischemic events of IS or TIA [13]. Whether the effective benefits of tranexamic acid can be offset by harm is still controversial. In addition, the present meta-analysis shows that the incidence of hydrocephalus and rebleeding has been significantly reduced compared to the control group in SAH patients. The reduction of hydrocephalus and rebleeding can decrease not only the occurrence of secondary surgery but also the secondary injury of neurons caused by intracranial haemorrhage. Therefore, the use of

tranexamic acid may have a beneficial effect on the improvement of long-term neurological function. However, the influencing factors of hydrocephalus and rebleeding are not only discussed, we needs to be more cautious about the results of data analysis [49].

The advantages of our study are as follows. First, we implemented a comprehensive search strategy in multiple databases. Second, we included 25 studies with large sample size of 20,146 participants. Third, all 25 studies included in the present analysis were RCTs. Therefore, the present meta-analysis may provide the highest-level evidence of evidence-based medicine on this topic.

Our study had some limitations. First, in the 25 included studies, treatment time (2 h to 7 days), and disease severity may have led to deviation of outcome data. Second, the follow-up time of each study was different, which may have affected the comparison between the tranexamic acid interventional group and the control group. Subacute, chronic, and late complications were more likely to be reported in studies with longer follow-up periods. Third, the location of intracerebral bleeding (intraventricular location, intraparenchymal location or both) is an independent predictor for the development of secondary hydrocephalus. The studies we included and our analysis did not control for this potential effect modifier. In addition, this association that tranexamic acid cannot improve the mRS score of ICH patients may also be confused by the location of ICH, affecting the functional outcome of three months.

## 5. Conclusion

Our study shows that tranexamic acid significantly reduced the risk of intracranial haemorrhage growth in patients with ICH and TBI without increasing the incidence of ischemic events when treating intracranial haemorrhage. Tranexamic acid intervention for SAH patients can significantly reduce the occurrence of hydrocephalus and rebleeding, and indirectly improve the quality of life of patients. ICH and TBI patients did not obtain the above benefits, which is the focus of our future research. In addition, we should also focus on the optimal time window and dose of tranexamic acid application to optimize patient treatment strategies.

## Supporting information

**S1 Checklist. PRISMA checklist.**
(DOCX)

**S1 Graphical abstract.** Schematic diagram of hemostatic mechanism of tranexamic acid (a) and chemical structure of tranexamic acid (b).
(TIF)

**S1 Fig. Flow chart describing the literature search.**
(TIF)

**S2 Fig. Risk of bias graph.** (a) The judgment of each bias risk item is expressed in percentage in all included studies. (b) Risk of bias summary.
(JPG)

## Acknowledgments

We would like to thank Editage (www.editage.cn) for English language editing.

## Author Contributions

**Data curation:** Chunhui Chen.

**Methodology:** Xiaodong Kang, Jianfeng Zhou, Roland Goldbrunner.

**Resources:** Xinyue Huang.

**Software:** Xiumei Guo.

**Supervision:** Linxing Wang, Pantelis Stavrinou.

**Validation:** Zhigang Pan, Lampis Stavrinou.

**Writing – original draft:** Yu Xiong.

**Writing – review & editing:** Yu Xiong, Shu Lin, Yuping Chen, Weipeng Hu, Feng Zheng.

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
