## [Decision Letter · Decision Letter 0]

13 Dec 2022

PONE-D-22-28422Efficacy and Safety of Tranexamic Acid in Cerebral Hemorrhage: A Meta-AnalysisPLOS ONE

Dear Dr. Zheng,

Thank you for submitting your manuscript to PLOS ONE. After careful consideration, we feel that it has merit but does not fully meet PLOS ONE’s publication criteria as it currently stands. Therefore, we invite you to submit a revised version of the manuscript that addresses the points raised during the review process.

We look forward to receiving your revised manuscript.

Kind regards,

Chinh Quoc Luong, MD., PhD.

Academic Editor

PLOS ONE

Journal Requirements:

Reviewers' comments:

Reviewer's Responses to Questions

**Comments to the Author**

1. Is the manuscript technically sound, and do the data support the conclusions?

Reviewer #1: Yes

Reviewer #2: Partly

2. Has the statistical analysis been performed appropriately and rigorously? 

Reviewer #1: Yes

Reviewer #2: No

3. Have the authors made all data underlying the findings in their manuscript fully available?

Reviewer #1: Yes

Reviewer #2: Yes

4. Is the manuscript presented in an intelligible fashion and written in standard English?

Reviewer #1: Yes

Reviewer #2: Yes

5. Review Comments to the Author

Reviewer #1: The authors of this manuscript conducted a meta-analysis of 27 studies, involving 22167 patients that sought to explore the hemostatic effect of tranexamic acid and its efficacy in improving neurological function in patients with acute intracerebral hemorrhage.

This manuscript offers insightful information and may be of interest to the readers of the journal provided the authors address certain queries and methodological concerns as outlined below:

1) “No significant effect was detected on the recovery of hematoma expansion and tranexamic acid use”. With regard to this statement the authors need to acknowledge the role of blood pressure control, cerebral amyloid angiopathy, chronic kidney disease and lifestyle (smoking, alcohol consumption) that may have confounded the reported associations. (PMID: 25156220, PMID: 36082246, PMID: 35870549).

2) “In addition, the present meta-analysis shows that the incidence of hydrocephalus has been significantly reduced compared to the control group in SAH patients". The location of intracerebral bleeding ( intraventricular location, intraparenchymal location or both) is an independent predictor for the development of secondary hydrocephalus (PMID: 16160424). The authors did not control for this potential effect modifier. Kindly, include this limitation in the revised discussion.

3) “Our results showed that tranexamic acid could significantly reduce the occurrence of hematoma growth in the brain, although no significant effect was detected on mRS”. This lack of association may also be confounded by ICH location that impacts functional outcomes at three months (PMID: 30287456, PMID: 24781084). Please acknowledge this limitation in the revised Discussion.

4) For outcome of more than 10 studies, funnel plot, with the use of Egger’s and Begg’s test was drawn. Please provide if trim and filling method was used to detect and adjust for publication bias (PMID: 31169736).

5. Please provide information on I2 squared statistic with regard to heterogeneity of the reported associations in the abstract.

Reviewer #2: Thank you for the opportunity to review this systematic review of tranexamic acid in ICH, SAH and TBI. I have a few comments as follows:

1) The terminologies used is unclear and overlapping. The authors state this systematic review on cerebral haemorrhage- but it appears this is more of intracranial haemorrhage. The common understanding of cerebral haemorrhage that it's intracerebral or intraparenchymal. There are a few contradictions here: TBI may not necessarily cause cerebral haemorrhage-in maybe subdural,extradural,subarachnoid, a combination of more than one or TBI may not have haemorrhage in case of diffuse axonal injury or in milder TBI-cerebral concussion.

Subarachnoid haemorrhage-are not considered intraparenchymal/intracerebral haemorrhage, though some may have ICH as well.

2) In line with point#1, ICH, SAH and TBI are all very different conditions, affecting different patient population, different pathophysiology, complications and prognosis. Furthermore, there's heterogeneity within each of the condition.

Therefore, trying to combine these 3 conditions into one systematic review may not be appropriate. They are managed by different specialities and readers may need to sieve through parts of systematic review to obtain the information of interest.

3) There are overlapping terminologies: haematoma growth and haematoma expansion are the same, how about rebleeding? how does one differentiate rebleeding from haematoma growth?

Not all studies will define haematoma expansion as > 6mL or >33%; how do the authors deal with different definitions?

4) Some studies included are the from the same RCT (i.e subgroup or secondary analysis) based on same dataset.

#7 Sprigg 2018, #11 Law 2021 and #18 Ovesen 2021 are all from TICH-2 trial

#31 and #35 Mahmood are from CRASH-3 trial

5) Search results

Despite searching for 3 different conditions, the number of studies are rather low. Baharoglu 2013 (https://www.cochranelibrary.com/cdsr/doi/10.1002/14651858.CD001245.pub2/full#CD001245-sec-0037) a systematic review on antifibrinolytic agents in SAH alone found 1045 studies.

No search strategies were given.

perhaps the authors should reexamine their search strategies.

6) risk of bias assessments

The authors may be overly lenient in assessing the risk of bias with majority of the studies had low risk of bias in most biases.

again a comparison with Baharoglu 2013 https://www.cochranelibrary.com/cdsr/doi/10.1002/14651858.CD001245.pub2/full#CD001245-sec-0037 showed that many of the same studies deemed to have low risk were assessed to have unclear or high risk by Baharoglu 2013.

perhaps the authors should justify why each risk of bias is assessed to be low with citation of text from the publication(s).

7) categorisation of safety outcomes

Why is mortality and hydrocephalus considered safety outcomes when they are most likely a result of the underlying condition rather than side effects of tranexamic acid?

8) page 25, line 412-414 tranexamic acid can also inhibit protein degradation into vasoactive peptides, ultimately reducing capillary permeability and increasing anti-vascular fragility (37). Moreover, tranexamic acid can enhance vascular permeability,

tranexamic acid both reduce and enhance vascular permeabilty-which is contradictory

9)Line 420 page 25 "Tranexamic acid blocks lysine binding sites and prevents fibrinogen from binding to fibrin molecules (46)"

actually TXA blocks activation of plasminogen and prevents binding of plasmin to fibrin; reference #46 also doesn't mention TXA mechanism.

10) page 17, 383-384:"In this meta-analysis, rebleeding occurred mainly in patients with ICH and TBI." I'm not sure the basis of this conclusion- the numbers in SAH seems similar to ICH

11) The grouping of outcomes: TIA/IS was grouped with epilepsy, but perhaps VTE and TIA/IS are more correlated-thrombotic complications and should be grouped together instead. Also seizures is better used than epilepsy as provoked seizures are not generally considered epilepsy

6. PLOS authors have the option to publish the peer review history of their article (what does this mean?). If published, this will include your full peer review and any attached files.

Reviewer #1: No

Reviewer #2: **Yes: **Zhe Kang Law

---

## [Author Response · Author response to Decision Letter 0]

8 Feb 2023

February 7, 2023

Plos One

Dear Editor Chinh , 

Thank you so much for your e-mail dated 14 Dec 2022, providing yours and the reviewers’ comments on our submission (Manuscript ID: PONE-D-22-28422R1) entitled “Efficacy and Safety of Tranexamic Acid in Intracranial haemorrhage: A Meta-Analysis”. Following your suggestions and the reviewer comments, our manuscript has been revised completely and re-submitted, along with the point-by-point responses to the comments. 

In the re-submitted manuscript, the revised parts have been marked in yellow. We believe that the revised manuscript can meet the publication standards of your journal.

Thank you again for your help in improving our manuscript.

Yours sincerely,

Feng Zheng

Professor and Director of Department of Neurosurgery,

The Second Affiliated Hospital,

Fujian Medical University

34 zhongshan North Road

Quanzhou 362000

Fujian Province

P.R. of China

Tel: + (86)13225985688 

Email: dr.feng.zheng@gmail.com

Reviewer 1 evaluation: The authors of this manuscript conducted a meta-analysis of 27 studies, involving 22167 patients that sought to explore the hemostatic effect of tranexamic acid and its efficacy in improving neurological function in patients with acute intracerebral hemorrhage.

This manuscript offers insightful information and may be of interest to the readers of the journal provided the authors address certain queries and methodological concerns as outlined below:

Q1:“No significant effect was detected on the recovery of hematoma expansion and tranexamic acid use”. With regard to this statement the authors need to acknowledge the role of blood pressure control, cerebral amyloid angiopathy, chronic kidney disease and lifestyle (smoking, alcohol consumption) that may have confounded the reported associations. (PMID: 25156220, PMID: 36082246, PMID: 35870549).

Response to the comment: Thank you for your comments. Yes, the role of blood pressure control, cerebral amyloid angiopathy, chronic kidney disease and lifestyle (smoking, alcohol consumption) may have confounded the reported associations, which has been acknowledged in the revised manuscript. Please see line 393-396 below. 

(line 393-396): Fourth, hematoma growth is also associated with blood pressure control, cerebral amyloid angiopathy, chronic kidney disease and lifestyle (smoking, alcohol consumption), which may have confounded the reported association.

Q2:“In addition, the present meta-analysis shows that the incidence of hydrocephalus has been significantly reduced compared to the control group in SAH patients". The location of intracerebral bleeding (intraventricular location, intraparenchymal location or both) is an independent predictor for the development of secondary hydrocephalus (PMID: 16160424). The authors did not control for this potential effect modifier. Kindly, include this limitation in the revised discussion.

Response to the comment: Thank you for your valuable suggestion. Yes, the location of intracerebral bleeding ( intraventricular location, intraparenchymal location or both) is an independent predictor for the development of secondary hydrocephalus, which has been included as limitation in the revised discussion. Please see line 385-389 below.

(line 385-389): Third, although the incidence of hydrocephalus has been significantly reduced compared to the control group in SAH patients, the location of intracerebral bleeding (intraventricular location, intraparenchymal location or both) as an independent predictor for the development of secondary hydrocephalus can not be controlled in the present analysis. 

Q3:“Our results showed that tranexamic acid could significantly reduce the occurrence of hematoma growth in the brain, although no significant effect was detected on mRS”. This lack of association may also be confounded by ICH location that impacts functional outcomes at three months (PMID: 30287456, PMID: 24781084). Please acknowledge this limitation in the revised Discussion.

Response to the comment: Thank you so much for your suggestions. This issue has been acknowledged as limitation in the revised Discussion. Please see line 389-393 below.

(line 389-393): In addition, although tranexamic acid could significantly reduce the occurrence of hematoma growth in the brain, no significant effect was detected on mRS. This lack of association may also be confounded by ICH location that impacts functional outcomes at three months. Future studies may be focused on this issue. 

Q4: For outcome of more than 10 studies, funnel plot, with the use of Egger’s and Begg’s test was drawn. Please provide if trim and filling method was used to detect and adjust for publication bias (PMID: 31169736).

Response to the comment: Thank you so much for your suggestions. Since we performed further analysis of the included studies based on disease type, there was no outcome of more than 10 studies in the revised manuscript. Therefore, funnel plot with trim and filling method is unnecessary to be drawn again in the re-submitted manuscript. Thanks for your advice and help.

Q5: Please provide information on I2 squared statistic with regard to heterogeneity of the reported associations in the abstract.

Response to the comment: Thank you very much for your suggestion. The information on the I2 squared statistics with regard to heterogeneity of the reported associations has been provided in the abstract. Please see line 83-91 below. 

(line 83-91): Twenty-five randomized controlled trials were included. Tranexamic acid significantly inhibited hematoma growth in intracranial hemorrhage (ICH) and traumatic brain injury (TBI) patients. (ICH: mean difference -1.76, 95%CI -2.78 to -0.79, I2=0%, P<.001; TBI: MD -4.82, 95%CI -8.06 to -1.58, I2=0%, P=.004). For subarachnoid hemorrhage (SAH) patients, it significantly decreased the risk of hydrocephalus (OR 1.23, 95%CI 1.01 to 1.50, I2=0%, P=.04) and rebleeding (OR, 0.52, 95%CI 0.35 to 0.79, I2=56% P=.002). There was no significance in modified Rankin Scale, Glasgow Outcome Scale 3-5, mortality, deep vein thrombosis, pulmonary embolism, or ischemic stroke/transient ischemic.

Reviewer 2 evaluation: Thank you for the opportunity to review this systematic review of tranexamic acid in ICH, SAH and TBI. I have a few comments as follows:

Q1: The terminologies used is unclear and overlapping. The authors state this systematic review on cerebral haemorrhage- but it appears this is more of intracranial haemorrhage. The common understanding of cerebral haemorrhage that it's intracerebral or intraparenchymal. There are a few contradictions here: TBI may not necessarily cause cerebral haemorrhage-in maybe subdural,extradural,subarachnoid, a combination of more than one or TBI may not have haemorrhage in case of diffuse axonal injury or in milder TBI-cerebral concussion. Subarachnoid haemorrhage-are not considered intraparenchymal/intracerebral haemorrhage, though some may have ICH as well.

Response to the comment: Thank you so much for your comments. Yes, the systematic review we present is more of intracranial haemorrhage. Following your idea, we replaced all the “cerebral haemorrhage” used in the title and text of the revised manuscript with “intracranial haemorrhage”. Thanks for your valuable suggestions.

Q2: In line with point#1, ICH, SAH and TBI are all very different conditions, affecting different patient population, different pathophysiology, complications and prognosis. Furthermore, there's heterogeneity within each of the condition.

Therefore, trying to combine these 3 conditions into one systematic review may not be appropriate. They are managed by different specialities and readers may need to sieve through parts of systematic review to obtain the information of interest.

Response to the comment: Thank you so much for your constructive comments. Following your suggestions, we performed further subgroup analysis based on 3 different disease types(intracranial hemorrhage, subarachnoid hemorrhage, traumatic brain injury) to systematically evaluate the role of tranexamic acid in these three different patient population. Please see line 253-334 below. 

(line 253-334): 

3.3 Outcomes

3.3.1 ICH

3.3.1.1 Hematoma growth

Four studies (7, 16-18) reported hematoma growth at the end of follow-up, and the pooled data showed that the hematoma growth rate in the tranexamic acid intervention group was lower than that in the control group (MD -1.78, 95%CI -2.78 to -0.79, I²=0%, P<.001; Fig 1A) for ICH patients.

3.3.1.2 mRS

Three studies (3, 17, 18) reported mRS scores at the end of follow-up, and pooled data showed that tranexamic acid had no significant benefit on the prognosis of patients with ICH (MD 0.08, 95%CI -0.31 to 0.47, I² = 0%, P=.69; Fig 1B).

3.3.1.3 mRS≤2

Three studies (3, 7, 17) reported an mRS score≤2 (non-disabling stroke patients) at the end of follow-up. No significant difference was detected between the tranexamic acid and control groups in ICH patients (OR 1.05, 95%CI 0.89 to 1.24, I²=0%; P=.58; Fig 1C).

3.3.1.4 Mortality

Four studies (3, 7, 17, 18) reported the number of deaths by the end of follow-up. There was no significant difference in the incidence of deaths between the tranexamic acid and control groups in ICH patients (OR 1.02, 95%CI 0.84 to 1.23, I²=0%, P=.84; Fig 1D).

Fig 1. Forest plot of the effect of tranexamic acid on hematoma growth (A), mRS (B), mRS≤2 (C), and Mortality (D) in patients with ICH. mRS: modified Rankin Scale.

3.3.2 SAH

3.3.2.1 GOS 3-5

Four studies (22, 26-28) reported a GOS score of 3–5 at the end of follow-up. There was no significant difference between the tranexamic acid intervention and control groups in SAH patients (OR 1.13, 95%CI 0.91 to 1.41, I²=0%, P=.26; Fig 2A). 

3.3.2.2 Mortality

Eight studies (19-21, 23-25, 27, 28) reported the number of deaths by the end of follow-up. There was no significant difference in the incidence of deaths between the tranexamic acid and control groups in SAH patients (OR 0.84, 95%CI 0.54 to 1.31, I²=57%, P=.83; Fig 2B).

3.3.2.3 DVT

Three studies (19, 25, 28) reported the occurrence of DVT at the end of follow-up. No significant difference was detected between the tranexamic acid intervention group and the control group in SAH patients (OR 1.08, 95%CI 0.51 to 2.30, I²=0%, P=.84; Fig 2C). 

3.3.2.4 IS or TIA

The occurrence of IS or TIA was reported in four studies (20, 25, 26, 28) at the end of follow-up. No significant difference was detected between the tranexamic acid and control groups in SAH patients (OR 1.20, 95%CI 0.83 to 1.72, I²=56%, P=.33; Fig 2D). 

3.3.2.5 Rebleeding

Rebleeding was reported in nine studies (19-23, 25-28) at the end of follow-up, and the data showed that the incidence of rebleeding in the tranexamic acid group was lower than that in the control group in SAH patients (OR 0.52, 95%CI 0.35 to 0.79; I²=56%, P=.002; Fig 2E).

3.3.2.6 Hydrocephalus

Four studies (24-26, 28) reported hydrocephalus at the end of follow-up, and pooled data showed that the incidence of hydrocephalus in the tranexamic acid group was lower than that in the control group in SAH patients (OR 1.23, 95%CI 1.01 to 1.50; I²=0%, P=.04; Fig 2F).

Fig 2. Forest plot of the effect of tranexamic acid on GOS 3-5 (A), Mortality (B), DVT (C), IS or TIA (D), Rebleeding (E) and Hydrocephalus (F) in patients with SAH. GOS: Glasgow Outcome Scale, DVT: deep vein thrombosis, IS: ischemic stroke, TIA: transient ischemic attack, SAH: subarachnoid hemorrhage.

3.3.3 TBI

3.3.3.1 Hematoma growth

Three studies (29, 33, 36) reported hematoma growth at the end of follow-up, and the pooled data showed that the hematoma growth rate in the tranexamic acid intervention group was lower than that in the control group (MD -4.82, 95%CI -8.06 to -1.58, I²=0%, P=.004; Fig 3A) for TBI patients.

3.3.3.2 Mortality

Six studies (13, 31, 32, 34, 35, 37) reported the number of deaths by the end of follow-up. There was no significant difference in the incidence of deaths between the tranexamic acid and control groups in TBI patients (OR 0.91, 95%CI 0.69 to 1.21, I²=0%, P=.52; Fig 3B).

3.3.3.3 PE

Three studies (13, 30, 35) reported the occurrence of PE at the end of follow-up. No significant difference was detected between the tranexamic acid intervention and control groups in TBI patients (OR 1.22, 95%CI 0.45 to 3.27, I²=65%; P=.70; Fig 3C).

3.3.3.4 IS or TIA

The occurrence of IS or TIA was reported in four studies (29, 30, 35, 37) at the end of follow-up. No significant difference was detected between the tranexamic acid and control groups in TBI patients (OR 0.81, 95%CI 0.51 to 1.30, I²=23%, P=.39; Fig 3D).

3.3.3.5 Seizures

Two studies (30, 35) reported the occurrence of seizures at the end of the follow-up. There was no significant difference between the tranexamic acid intervention and control groups in TBI patients (OR 1.11, 95%CI 0.92 to 1.36, I²=0%, P=.28; Fig 3E).

Fig 3. Forest plot of the effect of tranexamic acid on Hematoma growth (A), Mortality (B), PE (C), IS or TIA (D), and Seizures (E) in patients with SAH. PE: deep vein thrombosis, IS: ischemic stroke, TIA: transient ischemic attack, SAH: subarachnoid hemorrhage. 

Q3: There are overlapping terminologies: haematoma growth and haematoma expansion are the same, how about rebleeding? how does one differentiate rebleeding from haematoma growth? 

Response to the comment: Thank you for your comments. Yes, there are overlapping terminologies in our outcomes. Following your idea, a unified outcome indicator of hematoma growth was used in the revised manuscript. Besides, rebleeding was used in SAH patients, while hematoma growth was used in ICH and TBI patients. Please see line 254-259, line 296-300, and line 309-313 below. 

(line 254-259):

3.3.1 ICH

3.3.1.1 Hematoma growth

Four studies (7, 16-18) reported hematoma growth at the end of follow-up, and the pooled data showed that the hematoma growth rate in the tranexamic acid intervention group was lower than that in the control group (MD -1.78, 95%CI -2.78 to -0.79, I²=0%, P<.001; Fig 1A) for ICH patients.

(line 296-300):

3.3.2 SAH

3.3.2.5 Rebleeding

Rebleeding was reported in nine studies (19-23, 25-28) at the end of follow-up, and the data showed that the incidence of rebleeding in the tranexamic acid group was lower than that in the control group in SAH patients (OR 0.52, 95%CI 0.35 to 0.79; I²=56%, P=.002; Fig 2E).

(line 309-313):

3.3.3 TBI

3.3.3.1 Hematoma growth

Three studies (29, 33, 36) reported hematoma growth at the end of follow-up, and the pooled data showed that the hematoma growth rate in the tranexamic acid intervention group was lower than that in the control group (MD -4.82, 95%CI -8.06 to -1.58, I²=0%, P=.004; Fig 3A) for TBI patients.

Not all studies will define haematoma expansion as > 6mL or >33%; how do the authors deal with different definitions?

Response to the comment: Thank you for your comments. Yes, not all studies defined haematoma expansion as > 6mL or >33%. Of the 25 studies included, 3 studies (#Liu 2021, #Sprigg 2014, and #Sprigg 2018) defined the binary variable - haematoma expansion as “>6mL or >33% ”. 7 studies (#Arumugam 2015, #Liu 2021, #Sprigg 2014, #Sprigg 2018, #CRASH 2 2012, #Jokar 2017, #Safari 2021) reported the continuous variable - haematoma growth, and 1 (#Meretoja 2020) study collected the data of haematoma growth as a binary variable. Although the terms used are different, we believe that haematoma expansion and haematoma growth have the same meaning. Due to the absence of available data of binary variables and the fact that continuous variables and dichotomous variables cannot be combined to conduct a meta-analysis with one forest plot, we performed the analysis of the outcome of haematoma growth based on the detailed data of the continuous variable. Thanks again for your valuable comments.

Q4: Some studies included are the from the same RCT (i.e subgroup or secondary analysis) based on same dataset.

#7 Sprigg 2018, #11 Law 2021 and #18 Ovesen 2021 are all from TICH-2 trial

#31 and #35 Mahmood are from CRASH-3 trial

Response to the comment: Thank you for your comments. Yes, some studies included are the from the same RCT (i.e subgroup or secondary analysis) based on same dataset. Based on the considerations of data integrity , we included studies with the largest sample sizes (#7 Sprigg 2018 and #31 CRASH-3, which is #30 CRASH-3 in the revised manuscript), and other studies (#11 Law 2021, #18 Ovesen 2021 and #35 Mahmood) were therefore excluded. Please see the revised Table 1 below. 

Table 1. The characteristics of the 25 included studies.

Q5: Search results 

Despite searching for 3 different conditions, the number of studies are rather low. Baharoglu 2013 (https://www.cochranelibrary.com/cdsr/doi/10.1002/14651858.CD001245.pub2/full#CD001245-sec-0037) a systematic review on antifibrinolytic agents in SAH alone found 1045 studies. No search strategies were given. perhaps the authors should reexamine their search strategies.

Response to the comment: Thank you for your comments. Following your suggestion, the complete search strategy has been added in the revised manuscript. Furthermore, we confirmed our search results after repeated retrieval in the databases. The difference in search results between the paper by Baharoglu et al. and ours may be due to different search strategy. Namely, different search strategy may produce different search results. Furthermore, more studies eligible (25 studies) have been included in our analysis, compared to Baharoglu 2013`s (10 studies). Please see line 129-137. 

(line 129-137): The keywords used in this search strategy included "Randomized Controlled Trial" AND (“Subarachnoid Hemorrhage” OR “SAH” OR “Subarachnoid Hemorrhage, Aneurysmal” OR “Subarachnoid Hemorrhage, Spontaneous” OR “Perinatal Subarachnoid Hemorrhage”OR “Subarachnoid Hemorrhage, Intracranial” OR “Brain Injuries, Traumatic” OR “Trauma, Brain” OR “TBI” OR “Encephalopathy, Traumatic” OR “Posterior Fossa Hemorrhage” OR “Brain Hemorrhage” OR “Intracranial Hemorrhages”) AND (“AMCHA” OR “trans-4” OR “t-AMCHA” OR “AMCA” OR “Anvitoff” OR “Cyklokapron” OR “Ugurol” OR “KABI 2161” OR “Spotof ” OR “Transamin” OR “Amchafibrin” OR “Exacyl” OR "Tranexamic Acid"). 

Q6: risk of bias assessments

The authors may be overly lenient in assessing the risk of bias with majority of the studies had low risk of bias in most biases. again a comparison with Baharoglu 2013 https://www.cochranelibrary.com/cdsr/doi/10.1002/14651858.CD001245.pub2/full#CD001245-sec-0037 showed that many of the same studies deemed to have low risk were assessed to have unclear or high risk by Baharoglu 2013. perhaps the authors should justify why each risk of bias is assessed to be low with citation of text from the publication(s).

Response to the comment: Thank you for your comments. Following your suggestions, the risk of bias has been re-assessed and corrected in the revised manuscript. Please see S2 Figure below.

S2 Figure. Risk of bias graph. (a) The judgment of each bias risk item is expressed in percentage in all included studies. 

S2 Figure. Risk of bias graph. (b) Risk of bias summary

Q7: categorisation of safety outcomes Why is mortality and hydrocephalus considered safety outcomes when they are most likely a result of the underlying condition rather than side effects of tranexamic acid?

Response to the comment: Thank you for your comments. Yes, mortality and hydrocephalus are most likely a result of the underlying condition rather than side effects of tranexamic acid, which has been corrected in the revised manuscript. Please see line 202-296, line 306-310, and line 336-340 below. 

(line 269-275):

3.3 Outcomes

3.3.1 ICH

3.3.1.4 Mortality

Four studies (3, 7, 17, 18) reported the number of deaths by the end of follow-up. There was no significant difference in the incidence of deaths between the tranexamic acid and control groups in ICH patients (OR 1.02, 95%CI 0.84 to 1.23, I²=0%, P=.84; Fig 1D).\\

(line 281-285):

3.3.2 SAH

3.3.2.2 Mortality

Eight studies (19-21, 23-25, 27, 28) reported the number of deaths by the end of follow-up. There was no significant difference in the incidence of deaths between the tranexamic acid and control groups in SAH patients (OR 0.84, 95%CI 0.54 to 1.31, I²=57%, P=.83; Fig 2B).

(line 314-318):

3.3.3 TBI

3.3.3.2 Mortality

Six studies (13, 31, 32, 34, 35, 37) reported the number of deaths by the end of follow-up. There was no significant difference in the incidence of deaths between the tranexamic acid and control groups in TBI patients (OR 0.91, 95%CI 0.69 to 1.21, I²=0%, P=.52; Fig 3B).

Q8: page 25, line 412-414 tranexamic acid can also inhibit protein degradation into vasoactive peptides, ultimately reducing capillary permeability and increasing anti-vascular fragility (37). Moreover, tranexamic acid can enhance vascular permeability, tranexamic acid both reduce and enhance vascular permeabilty-which is contradictory

Response to the comment: Thank you for your comments. This issue has been revised. Please see line 348-353. 

(line 348-353): Tranexamic acid competitively inhibits the adsorption of plasminogen on fibrin by binding to the lysine binding site on fibrin, thereby inhibiting the activation of plasminogen and preventing the degradation of fibrinolytic proteins by fibrinolytic enzymes, thus achieving antifibrinolytic and hemostatic effects (42, 43). It can also increase collagen synthesis in fibrin clots, thereby increasing the strength and stability of clots and reducing bleeding (5, 44).

Q9: Line 420 page 25 "Tranexamic acid blocks lysine binding sites and prevents fibrinogen from binding to fibrin molecules (46)" actually TXA blocks activation of plasminogen and prevents binding of plasmin to fibrin; reference #46 also doesn't mention TXA mechanism.

Response to the comment: Thank you for your comments. The sentence mentioned, along with the reference, has been corrected. Please see line 355-356 below. 

(line 355-356): TXA blocks activation of plasminogen and prevents binding of plasmin to fibrin (47). 

47. Rodriguez-Garcia FA, Sanchez-Pena MA, de Andrea GT, Villarreal-Salgado JL, Alvarez-Trejo HJ, Medina-Quintana VM, et al. Efficacy and Safety of Tranexamic Acid for the Control of Surgical Bleeding in Patients Under Liposuction. Aesthetic Plast Surg. 2022;46(1):258-64.

Q10: page 17, 383-384:"In this meta-analysis, rebleeding occurred mainly in patients with ICH and TBI." I'm not sure the basis of this conclusion- the numbers in SAH seems similar to ICH

Response to the comment: Thank you for your comments. Yes, the numbers in SAH were similar to ICH. The incorrect sentence has been deleted in the revised manuscript.

Q11: The grouping of outcomes: TIA/IS was grouped with epilepsy, but perhaps VTE and TIA/IS are more correlated-thrombotic complications and should be grouped together instead. Also seizures is better used than epilepsy as provoked seizures are not generally considered epilepsy

Response to the comment: Thank you for your comments. Following your suggestion, VTE and TIA/IS have been grouped together. Besides, epilepsy has been replaced with seizures in the revised manuscript. Please see line 286-295, line 319-328, and line 329-334 below. 

(line 286-295): 

3.3.2 SAH

3.3.2.3 DVT

Three studies (19, 25, 28) reported the occurrence of DVT at the end of follow-up. No significant difference was detected between the tranexamic acid intervention group and the control group in SAH patients (OR 1.08, 95%CI 0.51 to 2.30, I²=0%, P=.84; Fig 2C).

3.3.2.4 IS or TIA

The occurrence of IS or TIA was reported in four studies (20, 25, 26, 28) at the end of follow-up. No significant difference was detected between the tranexamic acid and control groups in SAH patients (OR 1.20, 95%CI 0.83 to 1.72, I²=56%, P=.33; Fig 2D). 

(line 319-328): 

3.3.3 TBI

3.3.3.3 PE

Three studies (13, 30, 35) reported the occurrence of PE at the end of follow-up. No significant difference was detected between the tranexamic acid intervention and control groups in TBI patients (OR 1.22, 95%CI 0.45 to 3.27, I²=65%; P=.70; Fig 3C).

3.3.3.4 IS or TIA

The occurrence of IS or TIA was reported in four studies (29, 30, 35, 37) at the end of follow-up. No significant difference was detected between the tranexamic acid and control groups in TBI patients (OR 0.81, 95%CI 0.51 to 1.30, I²=23%, P=.39; Fig 3D).

(line 329-334): 

3.3.3 TBI

3.3.3.5 Seizures

Two studies (30, 35) reported the occurrence of seizures at the end of the follow-up. There was no significant difference between the tranexamic acid intervention and control groups in TBI patients (OR 1.11, 95%CI 0.92 to 1.36, I²=0%, P=.28; Fig 3E).

---

## [Decision Letter · Decision Letter 1]

20 Feb 2023

PONE-D-22-28422R1Efficacy and Safety of Tranexamic Acid in Intracranial Haemorrhage: A Meta-AnalysisPLOS ONE

Dear Dr. Zheng,

Thank you for submitting your manuscript to PLOS ONE. After careful consideration, we feel that it has merit but does not fully meet PLOS ONE’s publication criteria as it currently stands. Therefore, we invite you to submit a revised version of the manuscript that addresses the points raised during the review process.

We look forward to receiving your revised manuscript.

Kind regards,

Chinh Quoc Luong, MD., PhD.

Academic Editor

PLOS ONE

Journal Requirements:

Reviewers' comments:

Reviewer's Responses to Questions

**Comments to the Author**

1. If the authors have adequately addressed your comments raised in a previous round of review and you feel that this manuscript is now acceptable for publication, you may indicate that here to bypass the “Comments to the Author” section, enter your conflict of interest statement in the “Confidential to Editor” section, and submit your "Accept" recommendation.

Reviewer #1: All comments have been addressed

Reviewer #2: (No Response)

2. Is the manuscript technically sound, and do the data support the conclusions?

Reviewer #1: Yes

Reviewer #2: Yes

3. Has the statistical analysis been performed appropriately and rigorously? 

Reviewer #1: Yes

Reviewer #2: Yes

4. Have the authors made all data underlying the findings in their manuscript fully available?

Reviewer #1: Yes

Reviewer #2: Yes

5. Is the manuscript presented in an intelligible fashion and written in standard English?

Reviewer #1: Yes

Reviewer #2: Yes

6. Review Comments to the Author

Reviewer #1: The authors have adequately addressed my comments.

I am looking forward to seeing this manuscript in print.

Reviewer #2: Thank you for the revised manuscript that has addressed most of my comments. Just 2 minor points regarding Table 1

1) It still has haematoma expansion and haematoma growth in the outcome column. I understand the authors has standardised this by taking out haematoma expansion.

2) Kindly recheck the dosage of TXA, some appears incorrect. in Sprigg 2018, Sprigg 2014, Meretoja, Liu, Arumugam, CRASH-3 - the doses were 1g over 10 minutes followed by 1g over 8 hours - so total of 2g/day.

7. PLOS authors have the option to publish the peer review history of their article (what does this mean?). If published, this will include your full peer review and any attached files.

Reviewer #1: No

Reviewer #2: **Yes: **Zhe Kang Law

---

## [Author Response · Author response to Decision Letter 1]

21 Feb 2023

21 March, 2023

Plos One

Dear Editor Chinh , 

Thank you so much for your e-mail dated 20 Feb 2023, providing yours and the reviewers’ comments on our submission (Manuscript ID: PONE-D-22-28422R1) entitled “Efficacy and Safety of Tranexamic Acid in Intracranial haemorrhage: A Meta-Analysis”. Following your suggestions and the reviewer comments, our manuscript has been revised completely and re-submitted, along with the point-by-point responses to the comments. 

In the re-submitted manuscript, the revised parts have been marked in yellow. We believe that the revised manuscript can meet the publication standards of your journal.

Thank you again for your help in improving our manuscript.

Yours sincerely,

Feng Zheng

Professor and Director of Department of Neurosurgery,

The Second Affiliated Hospital,

Fujian Medical University

34 zhongshan North Road

Quanzhou 362000

Fujian Province

P.R. of China

Tel: + (86)13225985688 

Email: dr.feng.zheng@gmail.com

Reviewer 1 evaluation: The authors have adequately addressed my comments.

I am looking forward to seeing this manuscript in print.

Response to the comment: Thank you so much for your positive comments.

Reviewer 2 evaluation: Thank you for the revised manuscript that has addressed most of my comments. Just 2 minor points regarding Table 1.

Q1: It still has haematoma expansion and haematoma growth in the outcome column. I understand the authors has standardised this by taking out haematoma expansion.

Response to the comment: Thank you so much for your comments. Yes, we have standardised this by taking out haematoma expansion. And the issue mentioned in the outcome column has been corrected accordingly. Please see the revised Table 1 below. 

Table 1. The characteristics of the 25 included studies.

Q2: Kindly recheck the dosage of TXA, some appears incorrect. in Sprigg 2018, Sprigg 2014, Meretoja, Liu, Arumugam, CRASH-3-the doses were 1g over 10 minutes followed by 1g over 8 hours - so total of 2g/day.

Response to the comment: Thank you so much for your comments. Yes, according to your suggestion, the dose of TXA has been rechecked and corrected. Please see the revised Table 1 below. Thanks again for your great help to us.

Table 1. The characteristics of the 25 included studies.

---

## [Decision Letter · Decision Letter 2]

22 Feb 2023

Efficacy and Safety of Tranexamic Acid in Intracranial Haemorrhage: A Meta-Analysis

PONE-D-22-28422R2

Dear Dr. Zheng,

We’re pleased to inform you that your manuscript has been judged scientifically suitable for publication and will be formally accepted for publication once it meets all outstanding technical requirements.

Kind regards,

Chinh Quoc Luong, MD., PhD.

Academic Editor

PLOS ONE

Additional Editor Comments (optional):

Reviewers' comments:

Reviewer's Responses to Questions

**Comments to the Author**

1. If the authors have adequately addressed your comments raised in a previous round of review and you feel that this manuscript is now acceptable for publication, you may indicate that here to bypass the “Comments to the Author” section, enter your conflict of interest statement in the “Confidential to Editor” section, and submit your "Accept" recommendation.

Reviewer #2: All comments have been addressed

2. Is the manuscript technically sound, and do the data support the conclusions?

Reviewer #2: Yes

3. Has the statistical analysis been performed appropriately and rigorously? 

Reviewer #2: Yes

4. Have the authors made all data underlying the findings in their manuscript fully available?

Reviewer #2: Yes

5. Is the manuscript presented in an intelligible fashion and written in standard English?

Reviewer #2: Yes

6. Review Comments to the Author

Reviewer #2: All comments have been addressed.

I look forward to the publication of the article. thank you.

7. PLOS authors have the option to publish the peer review history of their article (what does this mean?). If published, this will include your full peer review and any attached files.

Reviewer #2: **Yes: **Zhe Kang Law

---

## [Editor Report · Acceptance letter]

23 Mar 2023

PONE-D-22-28422R2 

Efficacy and Safety of Tranexamic Acid in Intracranial Haemorrhage: A Meta-Analysis 

Dear Dr. Zheng:

I'm pleased to inform you that your manuscript has been deemed suitable for publication in PLOS ONE. Congratulations! Your manuscript is now with our production department. 

Kind regards, 

on behalf of

Dr. Chinh Quoc Luong 

Academic Editor

PLOS ONE